# Successful Interventional Management of Life-Threatening Bleeding after Oocyte Retrieval: A Case Report and Review of the Literature

**DOI:** 10.3390/medicina58111534

**Published:** 2022-10-27

**Authors:** Hiroyuki Tokue, Azusa Tokue, Yoshito Tsushima

**Affiliations:** Department of Diagnostic and Interventional Radiology, Gunma University Hospital, 3-39-22 Showa-Machi, Maebashi 371-8511, Gunma, Japan

**Keywords:** oocyte retrieval, in vitro fertilization, pseudoaneurysm, hemorrhage, transcatheter arterial embolization

## Abstract

Life-threatening bleeding after oocyte retrieval is unusual. We report a case of massive vaginal bleeding requiring transcatheter arterial embolization (TAE) after transvaginal US-directed follicle aspiration for oocyte retrieval and provide a brief review of cases in which the pseudoaneurysm of the injured artery was managed with a TAE approach. A 40-year-old woman presented massive vaginal bleeding after transvaginal ultrasonography-directed follicle aspiration for oocyte retrieval. Contrast-enhanced computed tomography revealed active bleeding from the uterine ostium. Transcatheter arterial embolization was performed for a pseudoaneurysm of the right pudendal artery to manage the hemorrhage. Potentially life-threatening bleeding should be recognized as a rare complication after oocyte retrieval to promptly establish the diagnosis and preserve the uterus.

## 1. Introduction

In vitro fertilization (IVF) is a common treatment worldwide for people who cannot conceive naturally. IVF is generally very safe, and those undergoing IVF rarely experience any health or pregnancy-related problems associated with the procedure [1]. Transvaginal ultrasonography (US)-directed follicle aspiration is a standard procedure for oocyte retrieval for most IVF cases, and life-threatening bleeding after oocyte retrieval has been reported in 0.8% of IVF cases [1]. Potential risk factors for bleeding after oocyte retrieval include coagulation disorders, multiple punctures, and adhesion of the ovary to other organs [2,3,4]. 

Here, we report a case of massive vaginal bleeding requiring transcatheter arterial embolization (TAE) after transvaginal US-directed follicle aspiration for oocyte retrieval and review the relevant literature. 

## 2. Case Presentation

A 40-year-old woman with a history of primary infertility for three years was transferred using emergency transportation from another hospital. She presented massive vaginal bleeding. Her vital signs were as follows: heart rate, 135 bpm; blood pressure, 80/65 mmHg; and respiratory rate, 25 breaths/min. The hemoglobin level was 6.1 g/dL. The patient became hemodynamically unstable and had hypotension. She had undergone transvaginal US-directed follicle aspiration to retrieve oocytes 6 h earlier at another hospital. This was her second attempt at oocyte retrieval.

The patient had no history of surgical intervention. Her left ovary with a chocolate cyst due to a history of endometriosis adhered to the uterus and was unsuitable for oocyte retrieval. Further, the right ovary was firmly fixed behind the middle uterine segment. Her physician assessed the difficulty in obtaining oocytes with conventional follicle aspiration through the vaginal fornix from the right ovary because the right ovary was malpositioned. In addition to conventional follicular aspiration through the vaginal fornix, transmyometrial oocyte retrieval was performed for the right ovary under US guidance using a 19-gauge aspiration needle. To perform transmyometrial puncture, an aspiration needle was inserted obliquely through the lower uterine segment to reach the right ovary. Color Doppler was used to check blood flow around the follicles and myometrium. Three oocytes were retrieved. 

10 min after the procedure, Pulsatile vaginal bleeding and pain persisted after the procedure. Gauze packing did not stop the vaginal bleeding. She was transferred to our hospital using emergency transportation because she became hemodynamically unstable and developed hypotension with gauze packing alone.

Contrast-enhanced computed tomography (CT) on admission revealed active bleeding from the uterine ostium suggesting a ruptured pseudoaneurysm (Figure 1). TAE was performed after transfusion with two units of packed red blood cells. TAE was performed 60 min after arrival at our hospital. Aortography showed active bleeding from the branch of the right pudendal artery (Figure 2A). A 1.8-Fr microcatheter was introduced near the bleeding point (Figure 2B). After performing TAE using gelfoam, the patient became hemodynamically stable. TAE lasted 30 min. No adverse events related to the procedure or rebleeding were noted. The patient was discharged from our hospital five days after the TAE. Since then, she has been undergoing fertility treatment.

IVF is an established treatment modality for infertility worldwide. In assisted reproduction, US-guided retrieval of oocytes through the vaginal fornix is now accepted as a safe and well-tolerated method involving a low overall complication rate [1]. Although bleeding is one of the complications of IVF resulting from US-guided transvaginal oocyte retrieval [2,3,4,5,6], it is generally insignificant and is often controllable with topical treatments, such as the application of pressure and/or topical hemostatic agents. Major vaginal hemorrhage due to vascular injury has been reported in 0.8% of IVF cases [1].

In our case, transmyometrial oocyte retrieval was performed in addition to conventional follicular aspiration through the vaginal fornix. This might have increased the risk of hemorrhage because vascular penetration with aspiration can cause myometrial trauma [7].

Very rare cases require TAE to stop bleeding associated with oocyte retrieval. We searched PubMed and Google Scholar to review the available literature (published until April 2022) on massive vaginal bleeding requiring TAE after transvaginal US-directed follicle aspiration for oocyte retrieval using the key terms “oocyte retrieval,’’ and “arterial embolization’’. There were six cases (including our case) requiring TAE after transvaginal US-directed follicle aspiration for oocyte retrieval due to pseudoaneurysm (Table 1). Moreover, there were two cases (case No.2 and our case) punctured through the myometrium. The pseudoaneurysm was found immediately or seven days after oocyte retrieval in three of these cases [3,4] and, surprisingly, during pregnancy in the remaining three cases [2,5,6]. It might be speculated that increased blood flow to pelvic organs and hormonal changes during pregnancy would enlarge the pseudoaneurysm and make it visible. Therefore, in IVF pregnancy cases, the presence of pseudoaneurysms should be checked not only after oocyte retrieval but also during pregnancy.

The management of pseudoaneurysms based on the fertility of the case may be controversial (Figure 3). TAE has some advantages. It is a safe, highly effective at stopping and preventing bleeding, minimally invasive technique that can be completed repeatedly. The disadvantage of TAE is the use of ionizing radiation, including the effect on fertility. However, the effects of exposure can be minimized by shortening the procedure time. Recanalization of a pseudoaneurysm requiring repeat TAE during pregnancy was noted in one of the six cases [6]. N-butyl-2-cyanoacrylate (NBCA) was used as the embolic material in three of the cases [2,3,6]. NBCA is a permanent liquid embolic material. None of them showed recurrence after TAE with NBCA, but subsequent gestation in women desiring future fertility has not been fully investigated. Gelfoam has a risk of pseudoaneurysm recanalization. However, we embolized the artery with a gelfoam considering future fertility. No reports of pregnancy and childbirth after the treatment of pseudoaneurysms after IVF are available. Further studies are required to determine the most appropriate embolic material.

In conclusion, it is important to recognize pseudoaneurysm as a rare complication after IVF to achieve a prompt diagnosis and enact uterus-preserving management for this potentially life-threatening disorder. Both the fertility specialist and general obstetrician/gynecologist should be aware of this to ensure better counseling of their patients and effective treatment.

## Figures and Tables

**Figure 1 medicina-58-01534-f001:**
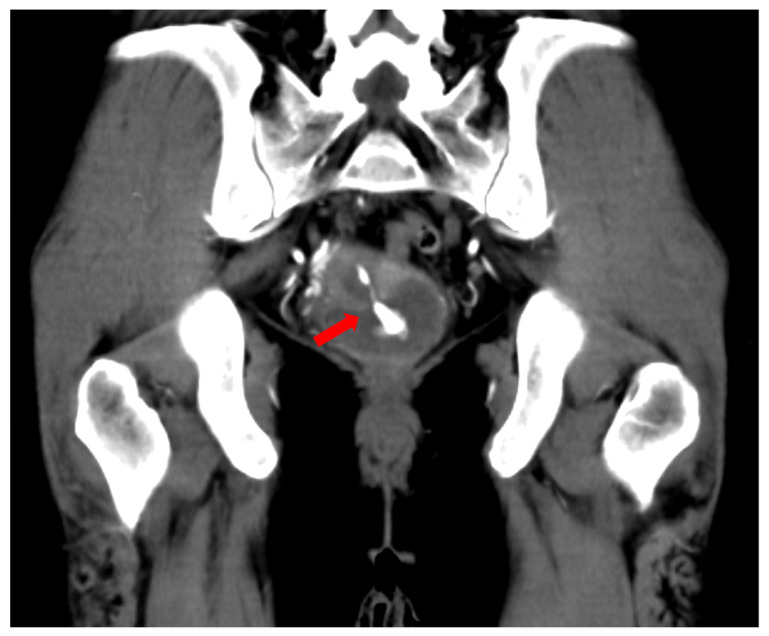
A case of a 40-year-old woman with massive vaginal bleeding. She underwent transvaginal US-directed follicle aspiration for oocyte retrieval 6 h earlier. Contrast-enhanced computed tomography (CT) on admission reveals active bleeding from the uterine ostium (arrow).

**Figure 2 medicina-58-01534-f002:**
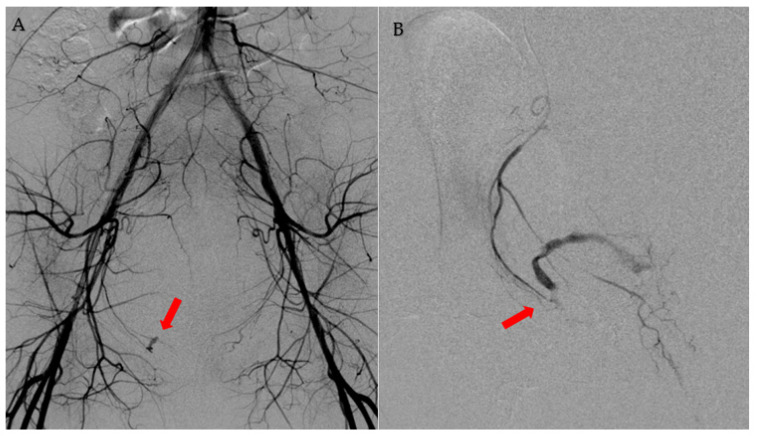
(**A**) Aortography showing active bleeding from the branch of the right pudendal artery (arrow). (**B**) A 1.8-Fr microcatheter is introduced near the bleeding point (arrow). TAE is performed using gelfoam.

**Figure 3 medicina-58-01534-f003:**
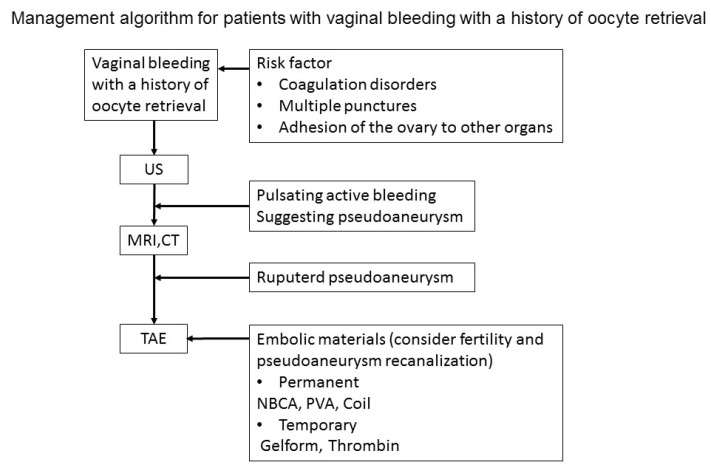
Management algorithm for patients with vaginal bleeding with a history of oocyte retrieval.

**Table 1 medicina-58-01534-t001:** TAE cases after transvaginal US-directed follicle aspiration for oocyte retrieval due to the presence of pseudoaneurysm.

Case	Authors	Age of the Patient (Years)	Past Medical History	Symptoms	The Day Pseudoaneurysm Was Observed	Diagnosis of Pseudoaneurysm	The Injured Artery	The Embolic Material
1	Bozdag et al. [2]	22	Nothing	Nothing	29 weeks of gestation (33 weeks after OR)	US, MRI	Left inferior pudendal artery	NBCA
2	Takeda et al. [3]	34	Four miscarriages	Vaginal bleeding	7 days after OR	US	Right uterine artery	NBCA
3	Kart et al. [4]	40	Mild factor VIII deficiency	Pain, pale, hypotension	Same day of OR	US	Right and left uterine arteries	PVA
4	Pappin and Plant [5]	37	Not reported	Vaginal bleeding	12 weeks of gestation (6 years after OR)	US, MRI	Left internal iliac artery	Coils, thrombin
5a *	Mulkers et al. [6]	35	Laparoscopy	Pain	19 weeks and 2 days of gestation	US, MRI	Left uterine artery	Microsphere, gelfoam, coils
5b *		35	Laparoscopy	Pain, vaginal bleeding	30 weeks of gestation	US, MRI	Left uterine artery	Glue
6	The present case	40	Nothing	Pain, hypotension, vaginal bleeding	Same day of OR	CT	Right pudendal artery	Gelfoam

TAE, transcatheter arterial embolization; OR, oocyte retrieval; NBCA, N-butyl-2-cyanoacrylate; PVA: polyvinyl alcohol. * 5a and 5b are the same patients. Recanalization of the pseudoaneurysm required repeated TAE during pregnancy.

## Data Availability

The data presented in the present study are available on request from the correspondent author.

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
