# Peer review of "Successful Interventional Management of Life-Threatening Bleeding after Oocyte Retrieval: A Case Report and Review of the Literature"

_medicina, 2022, doi:10.3390/medicina58111534_

Round 1
Reviewer 1 Report
Tokue and colleagues reported a case of massive vaginal bleeding (pseudoaneurysm of the right pudental artery) following transvaginal ultrasonography-directed follicle aspiration for oocyte retrieval. Although several cases of vaginal bleeding following oocyte retrieval have been reported in the literature, this is an interesting one and merits publication as it belongs to the few cases that were managed by transcatheter arterial embolization (TAE). The case report is accompanied by a brief literature review focusing on cases in which the same TAE-based uterine-preserving management approach was followed. Detailed comments on each section are provided below:
Abstract
"Life-threatening bleeding after oocyte retrieval is unusual.": This could be used as an introductory sentence before the case presentation.
The authors could also add that they provide a brief review of cases in which the pseudoaneurysm of the injured artery was managed with a TAE approach.
Introduction
The authors could add an introductory paragraph on reported complications of oocyte retrieval after transvaginal ultrasonography-directed follicle aspiration focusing on bleeding. They could refer to the epidemiology of bleeding (e.g., prevalence as estimated by existing literature) and potential risk factors (relevant patient history).
"IVF is generally very safe, and those undergoing IVF rarely experience any health or pregnancy-related problems associated with the procedure.": Could the authors cite appropriate references supporting this statement?
Case presentation
The case is clearly described. The procedure of oocyte retrieval, findings of contract-enhanced computed tomography (active bleeding of uterine ostium) and the procedure of TAE (including detection of the pseudoaneurysm of the right pudental artery by aortography before TAE) are all clearly presented. Was the patient checked for potential coagulation disorders?
Discussion
The authors focus on cases that were managed with TAE. It would be interesting to give readers a bigger picture by providing a figure proposing a management algorithm for patients with vaginal bleeding and a history of oocyte retrieval (based on current yet, of course, limited literature). Describing potential patient presentation, risk factors, diagnostic modalities and management approaches, the figure could focus on TAE and suggestions for appropriate embolic materials (as described in text), since the latter constitute the focus of this work.
Table
Could the authors add a column on how the pseudoaneurysm was evaluated in each reported case (e.g., ultrasonography, contrast-enhanced CT, apart from angiography)?
Author Response
Reviewer 1
Abstract
"Life-threatening bleeding after oocyte retrieval is unusual.": This could be used as an introductory sentence before the case presentation.
The authors could also add that they provide a brief review of cases in which the pseudoaneurysm of the injured artery was managed with a TAE approach.
We changed the abstract:
Life-threatening bleeding after oocyte retrieval is unusual. We report a case of massive vaginal bleeding requiring transcatheter arterial embolization (TAE) after transvaginal US-directed follicle aspiration for oocyte retrieval and provide a brief review of cases in which the pseudoaneurysm of the injured artery was managed with a TAE approach. A 40-year-old woman presented massive vaginal bleeding after transvaginal ultrasonography-directed follicle aspiration for oocyte retrieval. Contrast-enhanced computed tomography revealed active bleeding from the uterine ostium. TAE was performed for a pseudoaneurysm of the right pudendal artery to manage the hemorrhage. Potentially life-threatening bleeding should be recognized as a rare complication after oocyte retrieval to promptly establish the diagnosis and preserve the uterus.
Introduction
The authors could add an introductory paragraph on reported complications of oocyte retrieval after transvaginal ultrasonography-directed follicle aspiration focusing on bleeding. They could refer to the epidemiology of bleeding (e.g., prevalence as estimated by existing literature) and potential risk factors (relevant patient history).
"IVF is generally very safe, and those undergoing IVF rarely experience any health or pregnancy-related problems associated with the procedure.": Could the authors cite appropriate references supporting this statement?
We changed and added the Introduction
In vitro fertilization (IVF) is a common treatment worldwide for people who cannot conceive naturally. IVF is generally very safe, and those undergoing IVF rarely experience any health or pregnancy-related problems associated with the procedure [1]. Transvaginal ultrasonography (US)-directed follicle aspiration is a standard procedure for oocyte retrieval for most IVF cases, and life-threatening bleeding after oocyte retrieval has been reported in 0.8% of IVF cases [1]. Potential risk factors of bleeding after oocyte retrieval include coagulation disorders, multiple punctures, and adhesion of the ovary to other organs [2-4].
Case presentation
The case is clearly described. The procedure of oocyte retrieval, findings of contract-enhanced computed tomography (active bleeding of uterine ostium) and the procedure of TAE (including detection of the pseudoaneurysm of the right pudental artery by aortography before TAE) are all clearly presented. Was the patient checked for potential coagulation disorders?
She was not checked for potential coagulation disorders.
Discussion
The authors focus on cases that were managed with TAE. It would be interesting to give readers a bigger picture by providing a figure proposing a (based on current yet, of course, limited literature). Describing potential patient presentation, risk factors, diagnostic modalities and management approaches, the figure could focus on TAE and suggestions for appropriate embolic materials (as described in text), since management algorithm for patients with vaginal bleeding and a history of oocyte retrieval the latter constitute the focus of this work.
We added FIG3 (Management algorithm for patients with vaginal bleeding with a history of oocyte retrieval)
Table
Could the authors add a column on how the pseudoaneurysm was evaluated in each reported case (e.g., ultrasonography, contrast-enhanced CT, apart from angiography)?
We add a column of diagnosis of pseudoaneurysm.

Reviewer 2 Report
1. Information about initial immediate management of massive vaginal bleeding (first centre before transportation) ? only gauze packing for 6h ?
2. How long after arrival at your center was the embolization performed ?
3. How long did the embolization procedure last in this case?
4. How long after the follicular aspiration procedure was the hemorrhage diagnosed (first centre)?
5. Any sign of internal bleeding ?
6. “left ovary with a chocolate cyst adhered to the uterus” – history of endometriosis ?
7. Potential general complication of TAE (excluding the effect on fertility) ?
8. it is not clearly stated when and how the diagnosis of pseudoaneurysm was made in this case
9. What are the benefits of arterial embolization in these cases (hemorrhages after follicular aspiration) compared to other types of management-more discussion
1. Information about initial immediate management of massive vaginal bleeding (first centre before transportation) ? only gauze packing for 6h ?
2. How long after arrival at your center was the embolization performed ?
3. How long did the embolization procedure last in this case?
4. How long after the follicular aspiration procedure was the hemorrhage diagnosed (first centre)?
5. Any sign of internal bleeding ?
6. “left ovary with a chocolate cyst adhered to the uterus” – history of endometriosis ?
7. Potential general complication of TAE (excluding the effect on fertility) ?
8. it is not clearly stated when and how the diagnosis of pseudoaneurysm was made in this case
9. What are the benefits of arterial embolization in these cases (hemorrhages after follicular aspiration) compared to other types of management-more discussion
1. Information about initial immediate management of massive vaginal bleeding (first centre before transportation) ? only gauze packing for 6h ?
2. How long after arrival at your center was the embolization performed ?
3. How long did the embolization procedure last in this case?
4. How long after the follicular aspiration procedure was the hemorrhage diagnosed (first centre)?
5. Any sign of internal bleeding ?
6. “left ovary with a chocolate cyst adhered to the uterus” – history of endometriosis ?
7. Potential general complication of TAE (excluding the effect on fertility) ?
8. it is not clearly stated when and how the diagnosis of pseudoaneurysm was made in this case
9. What are the benefits of arterial embolization in these cases (hemorrhages after follicular aspiration) compared to other types of management-more discussion
1. Information about initial immediate management of massive vaginal bleeding (first centre before transportation) ? only gauze packing for 6h ?
2. How long after arrival at your center was the embolization performed ?
3. How long did the embolization procedure last in this case?
4. How long after the follicular aspiration procedure was the hemorrhage diagnosed (first centre)?
5. Any sign of internal bleeding ?
6. “left ovary with a chocolate cyst adhered to the uterus” – history of endometriosis ?
7. Potential general complication of TAE (excluding the effect on fertility) ?
8. it is not clearly stated when and how the diagnosis of pseudoaneurysm was made in this case
9. What are the benefits of arterial embolization in these cases (hemorrhages after follicular aspiration) compared to other types of management-more discussion
Author Response
Reviewer 2
- Information about initial immediate management of massive vaginal bleeding (first centre before transportation) ? only gauze packing for 6h ?
initial immediate management was only gauze packing for 6h.
She was transferred to our hospital using emergency transportation because she became hemodynamically unstable and developed hypotension with gauze packing alone.
- How long after arrival at your center was the embolization performed ?
TAE was performed 60 minutes after arrival at our hospital.
- How long did the embolization procedure last in this case?
TAE lasted 35 minutes
- How long after the follicular aspiration procedure was the hemorrhage diagnosed (first centre)?
10 minutes after the procedure, pulsatile vaginal bleeding and pain persisted
- Any sign of internal bleeding ?
No
- “left ovary with a chocolate cyst adhered to the uterus” – history of endometriosis ?
The patient had no history of surgical intervention. Her left ovary with a chocolate cyst due to a history of endometriosis adhered to the uterus and was unsuitable for oocyte retrieval.
- Potential general complication of TAE (excluding the effect on fertility) ?
Disadvantage of TAE is the use of ionizing radiation including the effect on fertility. However, the effects of exposure can be minimized by shortening the procedure time.
- it is not clearly stated when and how the diagnosis of pseudoaneurysm was made in this case
Contrast-enhanced computed tomography (CT) on admission revealed active bleeding from the uterine ostium suggesting a ruptured pseudoaneurysm (Fig. 1).
- What are the benefits of arterial embolization in these cases (hemorrhages after follicular aspiration) compared to other types of management-more discussion
TAE has some advantages. It is a safe and highly effective at stopping and preventing bleeding, minimally invasive technique that can be completed repeatedly.
No reports of pregnancy and childbirth after the treatment of pseudoaneurysms after IVF are available. Further studies are required to determine the most appropriate embolic material.
